# “Face(s)” of a PHACE(S) Syndrome Patient before and after Therapy: Particular Case Report and Review of Literature

**DOI:** 10.3390/children9121970

**Published:** 2022-12-15

**Authors:** Maria-Corina Stănciulescu, Florica Ramona Dorobantu, Eugen Sorin Boia, Marius-Călin Popoiu, Simona Cerbu, Rodica Heredea, Emil Radu Iacob, Anca Maria Cimpean, Borislav Dusan Caplar, Anca Voichita Popoiu

**Affiliations:** 1Department XI/Pediatric Surgery, Victor Babes University of Medicine and Pharmacy, 300041 Timisoara, Romania; 2Emergency Hospital for Children Louis Turcanu, 300041 Timisoara, Romania; 3Center of Expertise for Rare Vascular Disease in Children, Louis Turcanu Children Hospital, 300041 Timisoara, Romania; 4Department of Neonatology, Faculty of Medicine and Pharmacy, University of Oradea, 3700 Oradea, Romania; 5Department XV of Orthopaedics, Traumatology, Urology and Medical Imaging, Discipline of Radiology and Medical Imaging, Victor Babes University of Medicine and Pharmacy, 300041 Timisoara, Romania; 6Department V/Division of Clinical Practical Skills, Victor Babes University of Medicine and Pharmacy, 300041 Timisoara, Romania; 7Department of Microscopic Morphology/Histology, Victor Babes University of Medicine and Pharmacy, 300041 Timisoara, Romania; 8Angiogenesis Research Center Timisoara, Victor Babes University of Medicine and Pharmacy, 300041 Timisoara, Romania; 9Doctoral School in Medicine, Victor Babes University of Medicine and Pharmacy, 300041 Timisoara, Romania

**Keywords:** PHACE(S), infantile hemangiomas, propranolol therapy

## Abstract

A rare, uncommon disorder called PHACE(S) (P-posterior fossa anomalies, H-hemangioma, A-arterial anomalies, C-cardiac anomalies, E-eye anomalies, and S-sternal cleft) of unknown etiology was rarely reported. Children are susceptible to developing PHACE(S) syndrome from the moment they are born. It may be challenging for a physician to appropriately diagnose and treat children with PHACE due to the multifaceted nature of the disease and the extensive range of consequences that may be associated with it. A one-month-old newborn girl was admitted to hospital with extensive, multiple facial infantile hemangiomas, ulceration of the lower lip hemangioma-like lesion, cardiovascular, sternal, and neurological concomitant malformations. Five days following the initial application of the medication, systemic treatment with propranolol and topical treatment with silver sulfadiazine produced their first noticeable benefits. The lip ulceration was mostly healed and facial hemangioma started to regress. The regression continued under therapy and this effect persists for 6 months since Propranolol therapy ended. No cardiovascular or neurological clinical events have been registered during follow-up. The present case has three peculiarities: (1) high number of facial hemangiomas; (2) presence of subependymal cyst not yet reported in the literature associated with PHACE syndrome; and (3) lack of cardiovascular events during therapy knowing that these events frequently appear in PHACE syndrome patients.

## 1. Introduction

Since Pascual-Castroviejo published cases of infantile haemangioma coupled by cerebral abnormalities in 1978, the PHACE(S) has been recognized, under a variety of distinct names, as a syndrome that stands on its own: PHACE association, type II Pascual-Castroviejo syndrome, and complex of vascular anomalies (sternal malformations/vascular dysplasia and cutaneous haemangioma) are the conditions that are associated with this condition. Metry et al. provided the definitive consensus on the recognition of PHACES syndrome in 2008 [1]. The PHACE(S) syndrome includes the subset of children who are affected by large facial hemangiomas in addition to a series of other anomalies that compose the acronym of the syndrome as follows: posterior fossa anomalies (P), vascular abnormalities (such as large facial hemangiomas (H), arterial malformations (A), cardiac defects (C), and eye anomalies (E) [2,3,4]. 

PHACE syndrome is occasionally associated with endocrine dysfunctions [5], Moyamoya vasculopathy [6,7], hearing loss [8,9,10], or genetic syndromes [11]. According to the PHACE Syndrome International Clinical Registry and Genetic Repository (USA, 12), a small number of cases have been reported in Europe, limiting the prevalence of the disease and its specific features to a small case series [12].

Despite the fact that approximately 45 years have passed since the first description of the PHACE syndrome, its heterogeneity in clinical manifestations, controversies about associated diseases, and therapy response remains. According to Orphanet [13], this particular combination of symptoms is extremely rare, with fewer than a hundred cases registered into this database. Up to 350 cases have been documented in the literature to date but the total number of cases is unknown in this moment due to their partial report to national and international databases. Females are more likely to be affected than males (8:1) [13].

More and more evidence suggest that PHACE syndrome is a genetic disorder [14,15,16]. A clinical trial is currently underway to investigate the genetic disorder known as PHACE syndrome. Because of the rarity of PHACE, the clinical trial began in 2009 and is expected to last until 2030 [17]. Dental root anomalies, enamel hypoplasia [18,19], and lip hemangiomas [20] are also uncommonly reported in patients with PHACE syndrome.

Except for one paper reporting a late diagnosis of PHACE syndrome due to a subarahnoidian hemorrhage [21], this is the first case of PHACE syndrome reported in Romania early at childhood age. Furthermore, no reports of PHACE syndrome in East European countries have previously been published or registered to Orphanet, and no cases have been included in the Romanian Registry of Rare Diseases.

We thought it would be interesting to report our PHACE syndrome case because the subependymal cysts detection has not yet been reported to be associated with this condition, the absence of other significant visceral vascular anomalies in the context of the presence of extensive facial segmental hemangiomas, and lastly for a positive and persistent response to Propranolol without any side effects. 

## 2. Case Presentation

A female infant was born with pale skin areas on the face, bilaterally, on the genian region. This mark underwent significant changes in terms of color (red) intensity and surface area during her first week of postnatal life.

### 2.1. Clinical Data

The infant was admitted to the hospital at the age of one month with the following clinical features: two red masses on the genian area measuring 7 *×* 3 cm on the left cheek and 8 × 4 cm on the right. The lesions were located bilaterally on the genian area as well as on the upper and lower lip; they were soft and showed no signs of inflammation (Figure 1A–C). At the lower lip level, haemangiomatous-like lesions on the internal and external side of the lip can be seen, along with central ulceration and deep buccal fissures, making breastfeeding difficult and painful (Figure 1D,E). The patient had poor weight gain during the first month of life (250 g). The physical examination revealed a sternal cleft, which is more visible when crying (Figure 1E,F).

### 2.2. Cardiovascular and Neurological Evaluation

The patient was thoroughly investigated. ECG results showed a normal and regular sinus rhythm, a heart rate of 154 beats per minute, and a QRS axis of 135 degrees (right physiologic deviation, Figure 2A). The cardiac ultrasound revealed blood flow through a patent foramen ovale (Figure 2B), aortic arch left-deviated, from which right and left carotid arteries and left subclavian artery first arise, followed by the right subclavian artery, with the latter mostly showing a retrooesophageal pathway = arteria lusoria (Figure 2C); the presence of the arteria lusoria (Figure 2D).

Transcalvarial assessment—biventricular diameter 8.64 × 1.64 cm, lateral ventricles with symmetric anterior horns, according to cranial ultrasound. A hyperechogenic formation with 2–3 transonic formations, the largest of which was greater than 0.3 cm, was found in the left subependymal region. Subarachnoid space = 0.15 cm; longitudinal fissure = 0.2 cm. Based on this assessment, the final diagnosis was concluded to include left subependymal hemorrhage and subependymal cysts. Transcalvarial assessment was repeated every 3 months until spaces between the skull bones allowed this. No other congenital brain malformations were observed during repeated assessment.

All the above-mentioned observations of transcalvarial assessment are sustained by images from Figure 3. 

The clinical and paraclinical evaluation of the child lead to the diagnosis of PHACE(S) syndrome.

### 2.3. Therapy Management and Response

The patient’s treatment plan included the insertion of a naso-gastric tube for feeding, local applications of silver sulfadiazin to the labial ulceration, and oral systemic treatment with Propranolol (1 mg/kg/day on the first day, 2 mg/kg/day on the second day, and progressive adjustment of the beta-blocker dose based on weight gain). After 5 days of therapy, there was a significant reduction in the color intensity and surface of facial hemangiomas (Figure 4A–C). After two weeks, the nasogastric tube was removed. The child ate normally and gained weight normally. After 6 weeks of systemic Propranolol treatment, the local aspect improved, with a progressive decrease in the intensity of the color of the infantile haemangiomas and complete healing of the lower lip ulcer (Figure 4D–F). An ECG and a cardiac ultrasound were performed, and the beta-blocker dose was increased based on the patient’s current weight. Clinical and cardiological evaluations were performed three months after starting beta-blockers. Propranolol treatment was well-received by the patient. The clinical appearance of the child’s infantile haemangiomas improved as a result of an overall decrease in color intensity and surface area (Figure 4G–I). Depending on the infant’s current increasing weight, propranolol-based therapy was continued. After 6 months of systemic beta-blocker treatment, another clinical evaluation and cardiologic consult was performed. The patient’s overall progress was positive, and the cardiologic evaluation confirmed the child’s tolerance of Propranolol. The clinical appearance of infantile haemangiomas improves gradually over time (Figure 4J–L). Because of the natural evolution of infantile hemangiomas, it was decided to continue Propranolol treatment during the period of proliferation (the child being 7 months old). Regular checkups were performed every three months until the patient reached the age of two. Because infantile haemangiomas are in the maturation phase and relapse is theoretically very unlikely, patients are advised to discontinue beta-blocker treatment at the age of two. After two years and half, the child was evaluated during a routine inspection, and the hemangiomas were found to have lower marks (Figure 4M,N). Furthermore, the child was able to gain adequate weight (weight: 14 kg) and no severe sequels remained. The patient is currently undergoing periodic examinations to assess the status of the associated disorders (sternal cleft, arteria lusoria, subependymal cyst).

### 2.4. Case Peculiarities

Two facial hemangiomas measuring more than 5 cm in diameter, a sternal cleft, and an abnormal origin of the right subclavian artery are diagnostic clues of PHACE(S) syndrome after a thorough clinical and paraclinical evaluation of the child. Metry’s criteria classifies this case as sdr. PHACES because there are two infantile facial hemangiomas with diameters greater than 5 cm, as well as two more major criteria (arteria lusoria-aberrant origin of the right subclavian artery and sternal cleft). It is a complicated case with three major criteria for inclusion in PHACES and associated pathology. The uniqueness of the case stems from the presence of numerous facial hemangiomas with a total surface greater than 22 cm^2^ (7 × 3 + 8 × 4.5 + 3 × 2 = 63 cm^2^). Upper and lower lip ulcerated hemangioma-like lesions are also very rare for PHACE syndrome. The subependymal cyst was first described as being associated with PHACE syndrome here. Another feature of the case is the patient’s high tolerance to beta-blocker treatment. Patients with PHACES syndrome are more likely to experience beta-blocker side effects such as bradycardia and arterial hypotension. Finally, this is the first PHACE case reported in Romania and in the Eastern European region. 

## 3. Discussion

We report here a PHACE syndrome case early diagnosed in a one-month-old baby girl admitted to hospital for several facial hemangiomas, some of them ulcerated, and for slow weight gain. Two or more segmental facial hemangiomas with a total area more than 22 cm^2^ was considered one of the major diagnostic criteria for PHACE [1]. Two main peculiarities of our case must be highlighted related to facial hemangiomas: (1) the high number of facial hemangiomas including two upper and lower lip hemangiomas complicated with ulcerative lesions and (2) the extensive area covered by them (63 cm^2^). It was reported that cases with a high area of facial hemangiomas may have a higher rate of cardiovascular complications compared with cases having a smaller area of facial hemangiomas [22]. This clue is very important in the management of follow-up. At the age of two, the girl from the present study did not present any cardiovascular or neurological complications.

Lip hemangiomas (ulcerated or not) are rarely reported in PHACE syndrome [20,23,24]. A total number of 14 papers (out of 400) are reported in PubMed about the association of lip hemangiomas to PHACE syndrome [25]. Despite its rarity, the presence of lip hemangiomas seems to be associated with a high risk of recurrent late growth hemangiomas after the end of therapy and with a high risk of recurrent ulceration [26]. Ulcerated upper lip hemangiomas are frequently associated with eyelid ulcerated hemangiomas [27] or microphthalmia in PHACE syndrome [28]. Our case presented upper and lower lip ulcerated hemangiomas that were not associated with other facial ulcerated hemangiomas. Lip hemangiomas positively responded to topical and systemic therapy and had no recurrences at 2 years follow-up. No association with ocular hemangiomas or malformations was observed for the present case. Additionally, no enamel or dental root anomalies were reported during the child growth follow-up. 

Central nervous system structural malformations and cerebrovascular anomalies are commonly associated in the diagnosis of PHACE syndrome. Most patients with large area facial hemangiomas have associated nervous malformations from the most common one, hemispheric hypoplasia ipsilateral, to facial hemangioma, to the rarest one being corpus callosum agenesia [22]. Subependymal disorders are very rarely reported, with one paper describing gray matter heterotopia in the subependymal region [22]. 

Our case presented subependymal hemorrhage and subependymal cysts associated with PHACE syndrome. These were not described in the literature before. Subependymal hemorrhage regressed but subependymal cysts persists at 2 years follow-up with no other associated clinical manifestations. 

Last but not least, therapy response is an important step in the management of PHACE syndrome. Propranolol therapy response is variable and related to its effects on hemangioma regression, but is usually favorable, as happened in our case in a short period of time. Side effects of propranolol therapy such as bradycardia or hypotension were quite frequently reported for patients with PHACE syndrome, likely due to other vascular or cardiac malformations not fully investigated, but Olsen reported that no significant adverse effects have been registered for oral propranolol therapy in patients with PHACE syndrome [29]. Despite propranolol therapy’s wide acceptance for PHACE syndrome, some controversies still persist related to its long-term efficiency versus its side effects in children and its dosage for ulcerated versus non-ulcerated hemangiomas from PHACE syndrome according to the last guidelines recently published by VASCERN-VASCA working group [30].

There are more unknown than well-known data about PHACE syndrome in this moment. There were only four clinical trials in the world related to PHACE syndrome: one completed [31] focused on neurological disorders and the other three are in an actively recruiting state [17,32,33]. All of them were ruled in the United States of America. No official data about PHACE syndrome clinical trial ruled in Europe have been published.

One of the biggest controversies related to PHACE syndrome is if this condition is a genetic disease or not. Thus, in 2009, Stanford University, as the principal investigator, initiated the first clinical trial that focused on genetic analysis of PHACE syndrome [17]. This clinical trial is still open and will remain actively recruiting until 2030 because of the rarity of the disease. Two outcomes are in focus for this trial: (i) to establish a DNA and tissue bank and (ii) to determine candidate genes for PHACE syndrome using a genome-wide approach. The estimated number of enrolled cases was anticipated to be around 600 cases. In this moment there are no more than 400 cases worldwide, and thus any case report of PHACE syndrome seems to be very helpful. Inclusion criteria were based on the presence of extensive facial hemangiomas correlated with one or more other PHACE syndrome anomalies described at the beginning of the present paper. 

## 4. Conclusions

The first reported PHACE case from Romania and Eastern Europe has been described here as related to its clinical and therapeutic peculiarities. The lack of reporting such rare cases to the National Registry of Rare Disease may affect patient life quality by missing new symptoms that help the early diagnosis of such a disease. Follow-up of the patients for a long time is mandatory due to associated cardiovascular and neurologic pathologies.

## Figures and Tables

**Figure 1 children-09-01970-f001:**
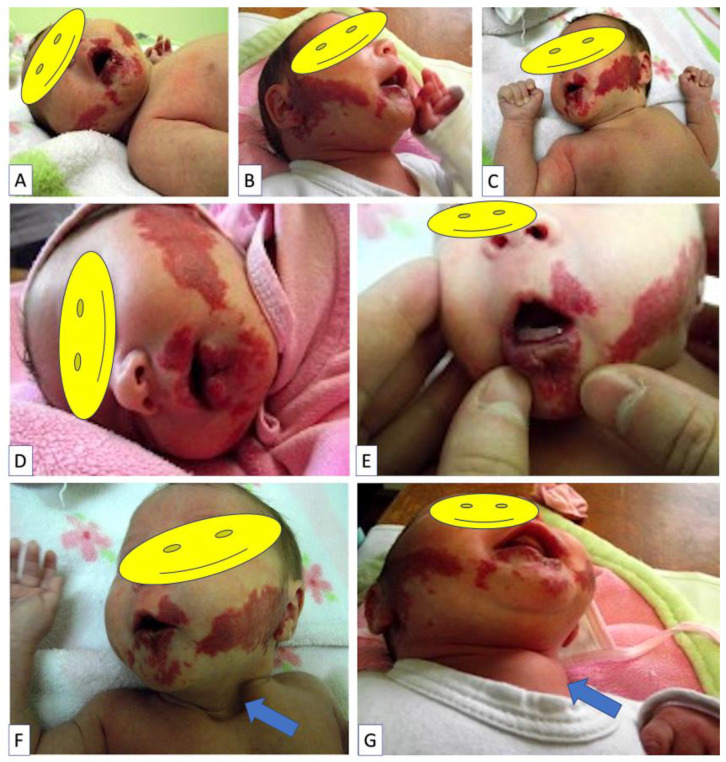
Clinical evaluation data of one month old baby girl admitted to hospital for multiple facial hemangiomas (**A**) distributed on the lower part of the face at mandibular region (right, (**B**) and left, (**C**)) and on upper (**D**) and lower (**E**) lip. Note a massive ulceration of lower lip hemangioma (**E**). A sternal cleft was observed in association with facial hemangiomas, (**F**) better visible when baby is crying (**G**).

**Figure 2 children-09-01970-f002:**
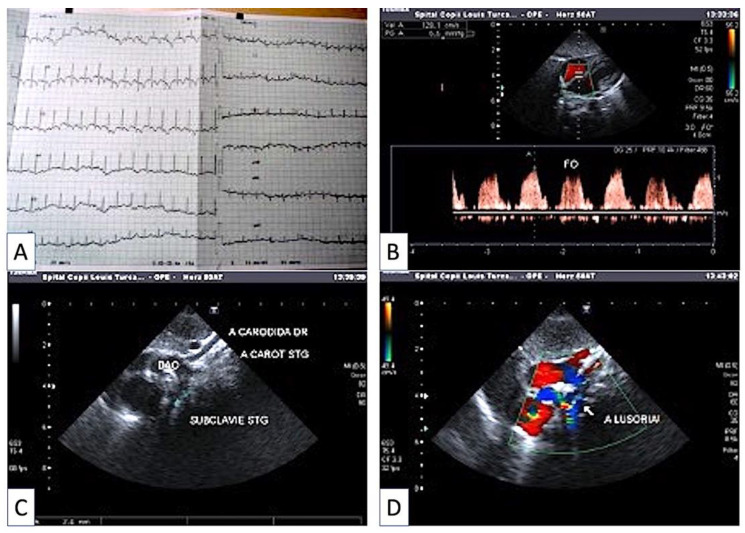
Cardiovascular assessment of the patient. Electrocardiogram appearance (**A**): RS regular 154 beats/min, axis QRS + 135 physiologically deviated to the right, otherwise normal electrical path. Blood flow presence through foramen ovale (**B**). Aortic arch to the left, from which emerge the right carotid artery, the left carotid artery, the left subclavian artery (**C**), and the right subclavian artery, which most often has a retroesophageal route = lusoria artery is observed (**D**).

**Figure 3 children-09-01970-f003:**
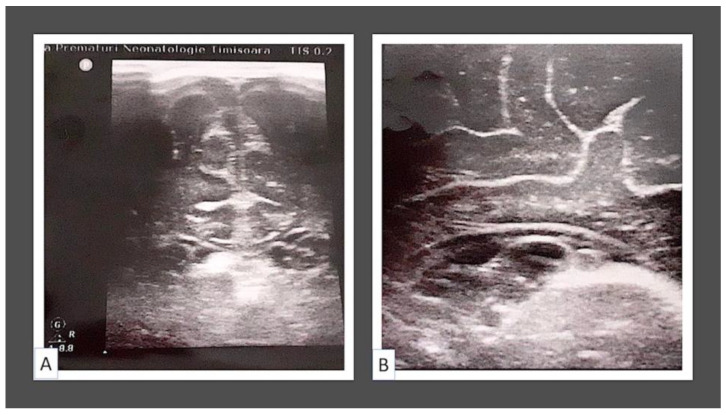
General overview of transcalvarial assessment (**A**) and detail of transcalvarial view of subependymal cysts (**B**). Note that there were not any other congenital brain malformations observed.

**Figure 4 children-09-01970-f004:**
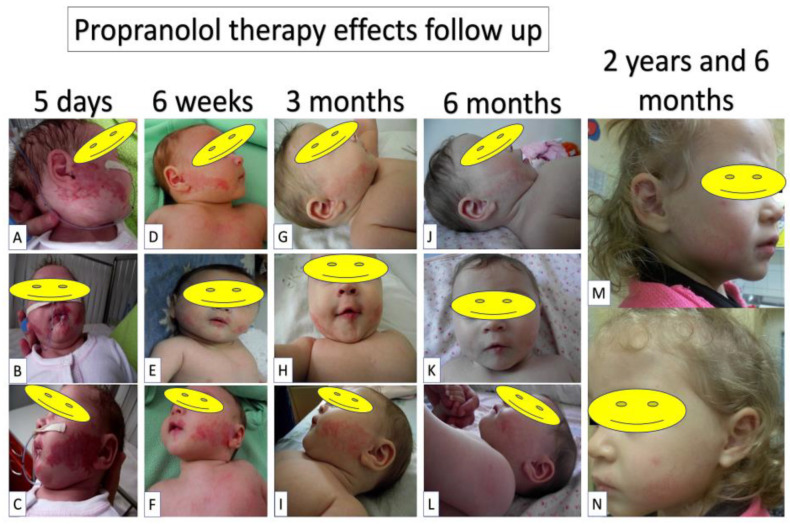
Extensive follow-up of facial hemangiomas regression as response to Propranolol therapy. After five days of therapy, the hemangiomas started to regress (**A**–**C**). Hemangioma regression continued and a reduced and paler area was observed at 6 weeks of therapy (**D**–**F**). After 3 months of therapy, lip hemangiomas were almost completely disappeared, and genian hemangiomas continued to regress (**G**–**I**). At 6 months, a high facial hemangioma regression continued to be observed (**J**–**L**). Therapy continued until 2 years of age. Six months after the end of Propranolol therapy, the facial hemangioma regression persisted and no recurrences were observed (**M**,**N**).

## Data Availability

All clinical and paraclinical data derived from the assessment of the patient and used here were stored into the archive of Louis Turcanu Children Hospital, Timisoara, Romania and may be available upon request.

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
