# Peer review of "“Face(s)” of a PHACE(S) Syndrome Patient before and after Therapy: Particular Case Report and Review of Literature"

_children, 2022, doi:10.3390/children9121970_

Round 1
Reviewer 1 Report
In this study the authors reported a case of PHACE syndrome. Some concerns and suggestions are listed as below:
Please provide the full name of PHACE in the manuscript.
In the part of introduction, please mention why this case is unique.
I wonder if this patient had congenital brain malformations. Please provide more evidence.
New comorbidities may be detected during follow-up. Did the authors perform any tests? How about long-term outcomes?
Author Response
Response to Reviewer 1
Dear Reviewer,
Thank you for your valuable review of our manuscript. Your observations are pertinent and welcome and we addressed them all, point by point. All changes requested by you, were highlighted in red in the revised version of the manuscript and explained in the present letter of response.
In this study the authors reported a case of PHACE syndrome. Some concerns and suggestions are listed as below:
Please provide the full name of PHACE in the manuscript.
The full name of PHACE is already stated in the manuscript (lines 59-61). We added now a detailed full name explanation of PHACE in the first sentence from Abstract section.
In the part of introduction, please mention why this case is unique.
The case peculiarities were mentioned in the last paragraph of the abstract. We also point out the most important peculiarities not yet reported in the literature (subependymal cyst) and a hugh area of segmental facial hemangiomas (about 63cm2 ) associated to PHACE. Being a very rare condition (about 100 reported in Orphanet but up to 400 cases in PubMed in the World)
I wonder if this patient had congenital brain malformations. Please provide more evidence.
No congenital brain malformations has been detected. We added images of transcalvarial assessement at the moment of diagnosis.
New comorbidities may be detected during follow-up. Did the authors perform any tests? How about long-term outcomes?
All these aspect has been mentioned in both case presentation and discussion sections.

Reviewer 2 Report
Dear Author,
Yours article is interesting but need some aspects to improve.
If title article include information about review literature, article should include
Follow information/design of trial
-inclusion criteria about article
Years of searching
Key word of searching
Author Response
Response to Reviewer 2
Dear Reviewer,
Thank you for your valuable review of our manuscript. Your observations are pertinent and welcome and we addressed them all, point by point. All changes requested by you, were highlighted in red in the revised version of the manuscript and explained in the present letter of response.
Dear Author,
Yours article is interesting but need some aspects to improve.
Thank you for your kind appreciation of our Case report of PHACE syndrome, a rare and less known disorder of childhood, usually underdiagnosed.
If title article include information about review literature, article should include
Present paper is a Case report and referred to a specified case but your comments was really valuable and gave me the idea to insert at the end of discussion section an overview for the only 4 clinical trials ruled in the world for PHACE syndrome. We detailed the largest one referring to inclusion criteria, outcomes and years as you suggest us.
Follow information/design of trial
-inclusion criteria about article
Years of searching
Key word of searching
There are more unknown than well-known data about PHACE syndrome in this moment. There were only 4 clinical trials in the world related to PHACE syndrome: one completed [31] focus on neurological disorders and other 3 in an actively recruiting state [17, 32, 33]. All of them are ruled in United States of America. No official data about PHACE syndrome clinical trial ruled in Europe has been published.
One of the biggest controversy related to PHACE syndrome is if this condition is a genetic disease or not. Thus, in 2009 Stanford University as principal investigator initiated the first clinical trial focused on genetic analysis of PHACE syndrome [17]. This clinical trial is still open and will remain actively recruiting till 2030 because of the rarity of the disease. Two outcomes are in focus for this trial: (i) to establish a DNA and tissue bank and (ii) to determine candidate genes for PHACE syndrome using a genome-wide approach. `The estimated number of enrolled cases was anticipated to be around 600 cases. In this moment there are no more than 400 cases worldwide and thus, any case report of PHACE syndrome seems to be very helpful. Inclusion criteria were based on the presence of extensive facial hemangiomas correlated with one or more other PHACE syndrome anomalies described at the beginning of the present paper.
